# The effect of a cognitive dual-task on the control of wheelchair propulsion

**Leon Salm** [1,2]*, **Lucas Schreff** [1], **Christian Benker** [1,2], **Rainer Abel** [1,2], **Roy Müller** [1,2,3]

**1** Department of Orthopedic Surgery, Klinikum Bayreuth GmbH, Bayreuth, Germany,
**2** Universitätsklinikum Erlangen, Friedrich-Alexander-Universität Erlangen-Nürnberg (FAU), Erlangen, Germany, **3** Bayreuth Center of Sport Science, University of Bayreuth, Bayreuth, Germany

* leon.salm@web.de

## Abstract

Movement analyses of wheelchair users are rarely integrated into clinical operations, although these are recommended to prevent pain and injury in the upper extremity. In addition, previous movement analyses in the laboratory do not include the possible influences of distractions that occur in everyday wheelchair use. We therefore transferred the dual-task method known from the motion analysis of walkers to the analysis of the propulsion behavior of wheelchair users and examined whether the changes resulting from the additional cognitive task also show up here. The 52 participants consisted of 38 manual wheelchair users (age=43.8±14.2 years; sex=11/27 f/m) and 14 novices (age=39.2 ± 15.4 years; sex=8/6 f/m). The participants propelled their wheelchairs on a test stand and movements of hand and wheel were recorded with a marker-based 3D motion-capturing system. The measurements were performed at preferred speed under single- and dual-task conditions. The Paced Auditory Serial Addition Test (PASAT) was used for the additional cognitive task. The participants propelled with a significantly higher frequency (p=0.001), had a shorter cycle- (p=0.001) and recovery time (p=0.001) and propelled with a lower push angle (p=0.045) under dual-task conditions. The distance between the hand and the pushrim was shorter (p=0.008) and the distance between the hand and the axis was significantly longer (p=0.004). The differences occurred predominantly in the group of manual wheelchair users. Significant differences in propulsion behavior were also found between the two groups under both single-task and dual-task conditions. The results indicate that dual-task conditions during wheelchair propulsion have an influence on spatiotemporal parameters similar to walking. Future movement analyses of wheelchair users should therefore consider the additional measurement under dual-task conditions in order to obtain more realistic results.

## 1 Introduction

Movement analyses of manual wheelchair users are rarely integrated into clinical operations. However, the analysis of propulsion behavior is extremely relevant. Unlike walking, using a wheelchair requires most of the power to be applied by the upper extremities, which places high muscular demands on them [1,2]. Therefore, poor performance in propulsion

**Data availability statement:** The datasets generated and analyzed during the current study

are available from Figshare at the DOI 10.6084/m9.figshare.26336968.

**Funding:** The author(s) received no specific funding for this work.

**Competing interests:** The authors have declared that no competing interests exist.

control very often leads to pain and injury in the upper extremity due to increased joint forces and an imbalance in muscular demands [3–5]. Most commonly, the pain occurs in the wrist and carpal tunnel (40% - 66%), the elbow (5% - 16%) and the shoulder (30% - 60%) [6]. It is therefore recommended that the wheelchair propulsion techniques of manual wheelchair users are regularly assessed and that they are then instructed to make any necessary changes [6].

However, even if movement analyses of manual wheelchair users are conducted, previous examinations in the laboratory do not include the possible influences of distractions that occur in everyday wheelchair use. But today's everyday life is full of distractions. Whether it's the advertisement on a passing cab that we glance after as we go through a traffic light, or the conversation we have on the way to work - we are constantly giving a portion of our attention to a variety of stimuli. This insight has also found its way into the scientific analysis of movements. These so-called dual-task situations are situations in which an action must be performed simultaneously with another task. The additional task increases the overall attention requirements and can lead to interference between the two tasks if the processing capacities are exceeded, resulting in reduced performance in one or both tasks [7,8]. Different options have been established for the additional cognitive task. In the past, multiple tests, including visual, auditory and arithmetic tasks, have been used for dual-task walking [9–12].

The dual-task movement analysis, i.e., the simultaneous execution of an additional cognitive task while walking and standing, already produced interesting results a few years ago [8,13–15]. For example, stride velocity was reduced and stride time variability was increased during dual-task overground walking compared to single-task walking [15,16]. More specifically, it is described that dual-task walking is associated with decreased stride length, suggesting that dual-task walking may negatively impact gait [15,17]. Regarding gait-specific parameters, Wang et al. described significant task effects on stride width and stride time [18]. Furthermore, a decrease in gait cadence and an increase in stride time were observed when walking with an additional cognitive task [15]. In addition, a decrease in swing time was also observed, whereas swing time variability was increased during dual-task walking [19]. Springer et al. also observed this effect, with the exception of the increase in swing time variability in young adults [20].

Now, not only ambulatory individuals are affected by distraction. Persons in manual wheelchairs are also confronted with a daily routine full of distractions. Thus, these distracting factors could also affect their movements while driving, i.e., their propulsion behavior. Even after extensive research, we hardly found any studies on the movement analysis of manual wheelchair users under dual-task conditions. However, this research is particularly interesting because propelling a wheelchair itself is a complicated motor task, more complicated and physically demanding compared to walking, that increases both the physical and cognitive workload of an individual [21]. As previous studies on wheelchair propulsion suggest that reducing the pushing frequency, increasing the push angle and using a semi-circular propulsion pattern or a pattern where the hand drops below the push frame towards the axis during the recovery phase are an effective way of propulsion [6,22], changes in these parameters are to be expected. We would therefore like to transfer the dual-task method known from the motion analysis of walkers to the analysis of the propulsion behavior of wheelchair users and find out whether a universal movement principle exists and the changes resulting from the additional cognitive task also show up here.

Based on these findings, we expect a shorter recovery time and a longer cycle time, as well as a smaller distance between hand and pushrim and a larger distance between hand and axle. We also hypothesize a smaller push angle and a lower speed and frequency for the dual-task propulsion compared to the single-task.

## 2 Methods

### 2.1 Participants

A total of 52 participants were included in this cross-sectional study. The sample size was calculated using an a priori power analysis for an ANOVA Repeated measures model conducted by means of G* Power 3.1.5 software, given the following input parameters: effect size: F = 0.2; alpha error probability: 0.05; power: 0.8. 33 of them were wheelchair-dependent, 19 were able to walk. The following inclusion criteria applied to the study participants: aged 18-65 years, no other orthopedic or neurological diseases that impair propulsion, no impairing asymmetry in movement/ severe scoliosis, currently pain-free. These inclusion criteria were met by all participants. The participants were divided into two groups (Table 1). The group of manual wheelchair users consisted of all wheelchair-dependent and five ambulatory but wheelchair-practiced participants (wheelchair basketball players, sports therapists), while the novice group consisted of 14 ambulatory and unpracticed participants. All participants provided written informed consent to participate in the study. The investigation was approved by the ethics review board of the Friedrich-Alexander University Erlangen-Nürnberg, Germany (23-20-B; 2023-02-14) and was in accordance to the Declaration of Helsinki. The recruitment period for this study extended from 2023-03-30 to 2024-06-12.

### 2.2 Procedure

The measurements took place in the gait and movement laboratory at Klinikum Bayreuth GmbH. Manual wheelchair users used their own adapted wheelchairs, ambulatory and inexperienced participants used a non-adapted active wheelchair (panthera; Model: S3 W: 42 cm).

The participants propelled their wheelchairs on a test stand in which each wheel rests on two rollers (double-roller) and all rollers can be moved independently of each other. They were secured by a protector system with four belts. The wheelchair propulsion movements were recorded with 10 infrared cameras (200 Hz) of a 3D marker-based system (Vicon, Oxford, UK).

Before the measurement began, the participants had the opportunity to do a few pushes to familiarize themselves with the driving conditions on the test stand and the experimental setup. In each case, the measurements took place at preferred speed without an additional cognitive task, then a sole execution of the Paced Auditory Serial Addition Test (PASAT) took place, after which the participants were measured again at preferred speed - but this time the additional cognitive task (PASAT) was completed during this time (dual-task). The PASAT, known from neurology, is a neuropsychological test to assess the capacity and speed of information processing as well as working memory and divided attention [23]. The participants heard numbers over a loudspeaker at 3-second intervals, then had to add the last two numbers mentioned and state the result immediately (Fig 1a). The test was ended after 10 results. In all

Table 1. Participants' characteristics separately for manual wheelchair users and novices.

| | Manual wheelchair users (N = 38) | Novices (N = 14) |
|---|---|---|
| age [years] | 43.8 ± 14.2 | 39.2 ± 15.4 |
| weight [kg] | 76.1 ± 16.6 | 76.2 ± 19.1 |
| arm length [cm] | 55.4 ± 4.9 | 55.8 ± 4.4 |
| sex [f/m] | 11/27 | 8/6 * |
| wheelchair dependent | 33 | 0 |

Note: Significant differences between manual wheelchair users and novices are marked by '*'.

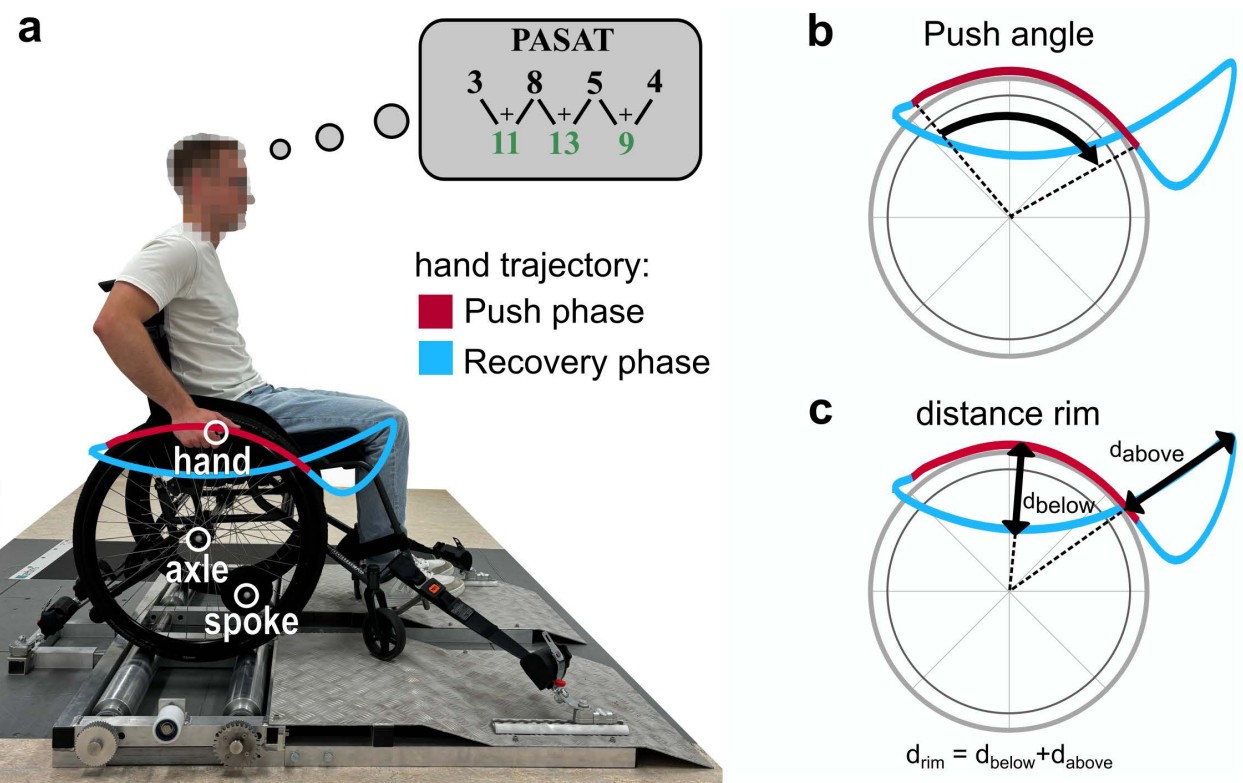

**Fig 1. Experimental setup and propulsion parameters.** The three motion capture markers were applied to the skin projection of the right second metacarpophalangeal joint, as well as to the axle and the spoke or the disk of the right wheel. In the thought bubble, the black numbers represent the PASAT numbers played by the loudspeaker and the green numbers represent the participant's results. The push angle was defined as the angle change of the pushrim during the push phase. The distance rim was calculated as the sum of the maximum hand distance above and below the pushrim.

test conditions with wheelchair propulsion, movements were recorded for 20 s. The recording was only started when the wheelchair was being driven and the participants had reached stable propulsion patterns. During the dual-task conditions, the measurement was started after the first two numbers of the PASAT had been stated.

## 2.3 Data processing and statistical analysis

In all measurements, the propulsion cycles were identified first. A propulsion cycle is divided into a push phase and a recovery phase. The push phase was defined as the cycle phase during which the hand is in contact with the wheel and pushes in a forward motion (detected as the acceleration of the wheelchair rear wheel). The Recovery phase was the period in which the hand did not propel the pushrim (detected as the deceleration of the wheelchair rear wheel). The acceleration and deceleration of the rear wheel was determined via double derivation of the path of the motion-capturing marker attached to the spokes. In all cases, only the right hand and the right wheel were analyzed.

Furthermore, the average velocity [24,25] (over the entire measurement time), the push frequency [24,26] and the cycle time were determined. We also assessed the time of the push phase and recovery phase, the push angle [22] (angle change of the pushrim during push phase) (Fig 1b), as well as the distance between the hand and the axle during recovery phase [22,25–27]. In addition, we introduced a variant to determine the maximum distance between the hand and the pushrim (for more detail see Fig 1c). This makes it possible to establish whether the

participant is more likely to keep their hand close to the pushrim or whether they are tending to release it. For the PASAT, the correctly named results out of 10 were counted.

Statistical analyses were performed with SPSS 20 (Chicago, IL, USA). To test normality of distributions, Kolmogorov-Smirnov tests were implemented for all spatiotemporal propulsion parameters as well as the cognitive performance (PASAT) during single- and dual-task (mean, standard deviation (SD)). Differences between manual wheelchair users and novices were assessed by an independent t-test. For the data that were not normally distributed (i.e., distance rim, distance axis and cognition during single- and dual-task) a Mann-Whitney-U-test was assessed. To evaluate the effect of a cognitive dual-task on the propulsion parameters, we performed a paired t-test for normally distributed parameters or a Wilcoxon test for not normally distributed parameters. Participant characteristics were compared using Pearson's Chi-square for gender and independent t-Tests for age, weight and arm length, separately for manual wheelchair users and novices. An alpha level of 0.05 was used for all statistical tests.

## 3 Results

The participants in the manual wheelchair users and novice group did not differ for age, weight and arm length (Table 1). However, the gender distribution differed in both groups (Table 1). For this paper, only mean values of spatiotemporal propulsion parameters were analyzed. S1 Table for standard deviation values of spatiotemporal propulsion parameters were provided as supplementary material.

### 3.1  Differences between single- and dual-task propelling a wheelchair

We compared the mean spatiotemporal propulsion parameters as well as the cognitive performance (PASAT) between single- and dual-task driving a wheelchair for the two groups to evaluate the influence of the additional cognitive task. Although the participants all had different propulsion patterns (Fig 2), we were able to detect significant changes in many parameters.

The manual wheelchair users group propelled with a significantly higher frequency during the additional cognitive task and had a shorter cycle time (Table 2). In addition, the recovery time was shorter and they propelled with a 3-degree lower push angle (Table 2). The distance between the hand and the pushrim was 15% shorter during the dual-task performance and the distance between the hand and the axis was significantly longer (Table 2). We did not observe any significant difference in velocity, push time and PASAT results compared to the single-task.

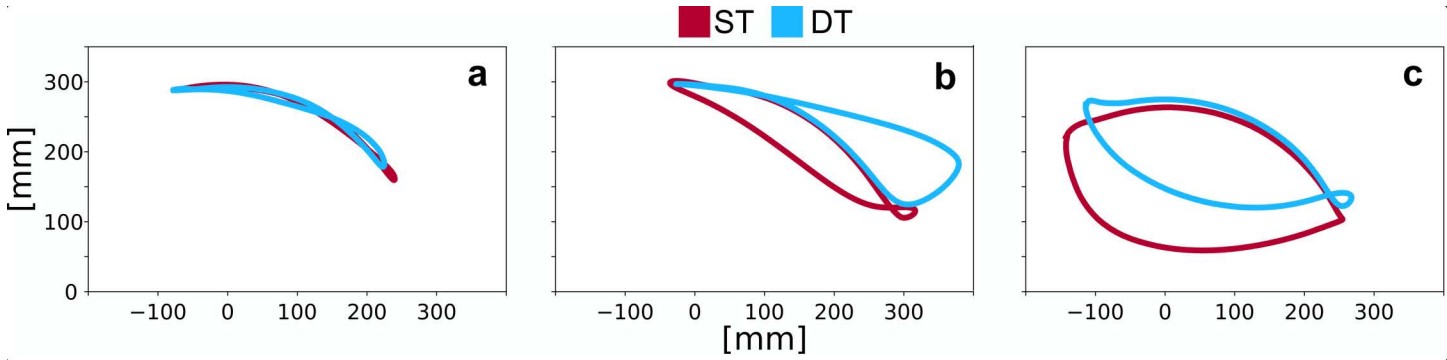

**Fig 2.  Propulsion patterns.** Examples of propulsion patterns according to Boninger et al. [27] during single- and dual-task propelling a wheelchair from our study are shown: (a) arc pattern (ARC); (b) single loop over pattern (SLOP); (c) semicircular pattern (SC). ST = single-task; DT = dual-task.

**Table 2. Comparison of the mean spatiotemporal propulsion parameters as well as the cognitive performance (PASAT) during single- and dual-task driving a wheelchair.**

| | Single-Task | | Dual-Task | | ST vs DT (p-value) | |
|---|---|---|---|---|---|---|
| | Manual wheelchair users | Novices | Manual wheelchair users | Novices | Manual wheelchair users | Novices |
| Velocity [m/s] | 0.81 ± 0.32 | 0.53 ± 0.26 * | 0.81 ± 0.35 | 0.55 ± 0.28* | 0.932 | 0.597 |
| Frequency | 54.5 ± 7.9 | 54.1 ± 13.9 | 59.1 ± 11.4 | 57.0 ± 11.6 | 0.001 | 0.152 |
| Cycle time [s] | 1.12 ± 0.15 | 1.17 ± 0.27 | 1.05 ± 0.18 | 1.10 ± 0.25 | 0.001 | 0.077 |
| Push time [s] | 0.42 ± 0.08 | 0.43 ± 0.09 | 0.41 ± 0.08 | 0.45 ± 0.14 | 0.399 | 0.521 |
| Recovery time [s] | 0.70 ± 0.10 | 0.74 ± 0.20 | 0.64 ± 0.11 | 0.65 ± 0.14 | 0.001 | 0.002 |
| Push angle [deg] | 65,6 ± 17.3 | 47,1 ± 9,5 * | 62,8 ± 18,2 | 46.8 ± 10.6* | 0.045 | 0.845 |
| Distance rim [cm] | 12.4 ± 7.8 | 6.6 ± 6.5 * | 10.6 ± 8.1 | 6.8 ± 7.6* | 0.008 | 0.925 |
| Distance axis [cm] | 24.1 ± 7.1 | 27.7 ± 2.1 | 26.1 ± 5.4 | 27.9 ± 1.8 | 0.004 | 0.300 |
| Cognition | 7.68 ± 2.33 | 9.14 ± 0.95 | 7.87 ± 2.24 | 9.07 ± 1.21 | 0.382 | 0.819 |

Note: significant differences between manual wheelchair users and novices are marked by '*'. The parameters of the single-task propulsion and the single-task PASAT are summarized under "single-task" for the sake of overview.

In the novice group, we only found a significant difference in the recovery time. Here, the participants propelled their wheelchair during the additional cognitive task with a 12% shorter recovery time (Table 2) compared to single-task propelling. We found no significant differences between any of the other parameters.

## 3.2 Differences between manual wheelchair users and novices

To identify the differences between manual wheelchair users and novices, we compared their spatiotemporal propulsion parameters as well as the cognitive performance (PASAT). During the single-task, the novices propelled their wheelchair significantly slower, had a 19-degree lower push angle and drove with a 47% smaller mean distance to the pushrim (Table 2). No significant differences were found between the frequency, the cycle time, the push/ recovery time, the distance between hand and axis in the recovery phase and the cognitive performance.

Under dual-task conditions, we found significant differences between the velocity, the push angle and the distance to the pushrim. More precisely, the participants in the novice group were significantly slower than the manual wheelchair users group and used a 16-degree smaller push angle. Their mean distance to the pushrim was 36% shorter compared to the manual wheelchair users (Table 2). No significant difference was found between the other parameters.

## 4 Discussion

The results indicate that dual-task conditions during wheelchair propulsion have an influence on spatiotemporal parameters similar to walking. Surprisingly, most of the significant changes between single- and dual-task were found mainly in manual wheelchair users and not in novices, although this is probably due to the different pushing behavior of the two groups.

### 4.1 Effect of a cognitive dual-task on propulsion behavior

When comparing the spatiotemporal propulsion parameters of single-task and dual-task, it is noticeable that the participants in the manual wheelchair users group propelled with a higher frequency and correspondingly shorter cycle time, as well as using a smaller push angle and

holding the hand closer to the pushrim (shorter hand-pushrim distance and longer hand-axis distance) (Table 2). All these changes could indicate that the participants focused more on the additional cognitive task and thus chose a safer way of propulsion. In fact, they propelled the wheels more often with a shorter and smaller push and kept their hand in the reach of the pushrim to maintain control and be able to quickly give the next push. In both groups, the recovery time was significantly shorter under dual-task conditions (Table 2). This can also be explained by a reaction to the reduced attention due to the additional cognitive task. After being pushed, the participants tend to pick up the pushrim again as quickly as possible. The fewer differences in the spatiotemporal propulsion parameters in the novice group under dual-task conditions could be caused by the smaller sample size of this group, the greater scatter in the parameters (Table 2), as well as by the fact that they already had a worse and therefore less cognitively demanding performance in propulsion control under single-task conditions and therefore there is less room for a significant worsening through dual-task conditions. In addition, experienced wheelchair users may be more able to adapt their propulsion style to any additional cognitive demands than inexperienced users.

## 4.2  Comparison with dual-task walking

Analyses of spatiotemporal propulsion parameters during walking under dual-task conditions show a shorter stride length [15,17] and a reduced stride velocity [15–17]. While we did observe a smaller push angle (Table 2), which can be seen as equivalent to the stride length at walking, we did not observe a significant change in velocity in our study. This may be related to the fact that our participants were measured on a test stand and therefore the visual environment did not change. In most studies on dual-task walking, participants walked across the room and not on the spot [13,17,19,20], which gave them feedback on their speed at all times. In addition, a decrease in swing time was observed during dual-task walking [19]. If you consider the recovery phase when driving a wheelchair to be equivalent to the swing time when walking, we were also able to observe this effect in our study (Table 2). Furthermore, a decrease in gait cadence was observed during dual-task walking [15]. In our study, the frequency when propelling the wheelchair increased (Table 2). The cycle time determined in our measurements can be compared with the stride time for walking. Here, an increase in stride time was observed under dual-task conditions [15], while the cycle time decreased in our study (Table 2). An increase in the variability of stride time/cycle time was observed in dual-task walking [15] as well as in our study (S1 Table, Supporting information). However, this change was not significant here. The changes we observed in the distances between the hand and axis and the hand and pushrim during the dual-task situation can be compared with the minimum toe clearance (MTC) or minimum foot clearance (MFC) during walking, as all represent the distance to the element providing safety. However, only significant differences in the variability of the MTC were found when walking [14], while changes in the mean value were not consistent [28].

## 4.3  Effect of experience on propulsion behavior

When comparing the manual wheelchair user group with the novice group, it is noticeable that the novices propel more slowly and with a lower push angle under both single-task and dual-task conditions (Table 2). These two differences could be explained by the different experience in propelling. For example, through wheelchair training, which many wheelchair users receive, or simply by optimizing their own efficiency, the wheelchair users may learn to use as large an angle as possible for propulsion and also dare to move faster. The smaller push angle among the novices was also found in earlier studies [29]. In addition, the novices had a smaller

distance between their hand and the pushrim under single-task conditions than the manual wheelchair users (Table 2). It is plausible that the novices are less confident in releasing their hand from the pushrim, for example to gain momentum, as they try to maintain control of the pushrim. Many unpracticed users are afraid of releasing the rim and choose propulsion patterns which minimize the time away from the pushrim and keep the hand close to the rim at all times [25]. Furthermore, most of the manual wheelchair users were measured in their own adapted wheelchairs, while the novices were provided with a wheelchair that was not adapted. It is possible that the propulsion behavior of the novices would have been different if they had also been measured in a wheelchair adapted to their individual body.

### 4.4 Calculation of hand-axis and hand-pushrim distance

Earlier studies on the movement analysis of wheelchair propulsion used the propulsion patterns according to Boninger et al. [27] to analyse the movement of the hand in relation to the pushrim. The semicircular and double-loop patterns are considered to be the most efficient [30]. However, as it is difficult to objectify the analysis of these patterns, the inter-rater reliability is questionable. The recommendation in the clinical practice guideline [6], which states that the hand should be allowed "to drift down naturally, keeping it below the pushrim when not in actual contact with that part of the wheelchair" [6], can be assessed more objectively. To quantify this, earlier studies calculated the distance of the hand marker from the axis at the closest point [22]. However, if a propulsion pattern is used in which the hand is also partially returned above the pushrim during the recovery phase (e.g., in Fig 2b and Fig 1), the difference is not represented using this method. In this calculation, the dual-task trajectory in Fig 2b would have exactly the same distance to the axis as the arc pattern in Fig 2a, even though they each have different biomechanical effects. We therefore introduced a new method for calculating the relation between hand and pushrim. For the parameter "distance rim" we calculate the sum of the maximum distances of the hand above and below the handrim in the recovery phase (Fig 1c). This allows us to determine not only whether the hand is returned below the pushrim, but also whether it is partially or fully returned above the pushrim.

Especially when combining "distance rim" with "distance axis", objective numerical values can be used to draw fairly accurate conclusions about the propulsion patterns. If, for example, the distance axis is similar to the radius of the wheel and the distance rim approaches zero, it can be assumed that this is an arc pattern (Fig 2a). If the distance axis is approximately equal to the radius and the distance rim is greater than zero, this indicates a single loop over pattern (Fig 2b). And if the distance axis is rather small and the distance rim is about the same size, we can assume a semicircular pattern (Fig 2c). This method is therefore a more objective and accurate way of analysing the relation between hand and pushrim.

### 4.5 Limitations

This study has potential limitations. Firstly, the group of novices only consists of 14 participants, which could have led to no significant changes being observed for some parameters under dual-task conditions in this group. The unequal group sizes could make a comparison between them less equitable, even if the effect sizes (Cohen's d) were around 0.8 except for "distance rim" under dual-task conditions. However, unlike the other parameters, which showed significant differences, "distance rim" was not normally distributed. In addition, there were differences in the gender distribution between the two groups, which could have had an impact on the results if one assumes that men and women propel a wheelchair differently. Secondly, only the PASAT was used to increase the cognitive workload. However, since the

PASAT may not be a pure measure of general information processing capacity [31], the use of and comparison with other distracting tasks, e.g., visual or auditory tasks, would have enabled more comprehensive results. Thirdly, the measurements were carried out in a movement laboratory due to the use of marker-based motion capturing, so that it cannot be ruled out that real-world conditions might additionally influence the results. In addition, most participants in the group of manual wheelchair users were measured in their own and adapted wheelchairs, while the novices were measured in a non-adapted wheelchair. This difference could limit a comparison between the two groups.

### 4.6 Perspectives

Our results show that there are differences between single- and dual-task propelling a wheelchair. Thus, travelling with a manual wheelchair binds attention, as already observed with walking. Future movement analyses of wheelchair users should therefore consider the additional measurement under dual-task conditions in order to obtain more realistic results. Dual-task measurements are suitable for more realistic results in the laboratory setting. However, it is most realistic to measure outside the laboratory. The recently investigated detection of spatiotemporal propulsion parameters using inertial measurement units (IMUs) opens up new possibilities for this [32]. This method also makes it possible to investigate the influence of an additional task on propelling a wheelchair on different surfaces or at different speeds and gradients [24,33]. In addition, future studies could simulate everyday distractions even better, for example by performing the measurements while the test subjects are making a phone call or are driving through a visually stimulating environment.

Another relevant study would be to divide the participants into different subgroups and compare the changes in the individual groups with each other. A subdivision could be made, for example, according to the level of paralysis, the time spent in a wheelchair or the activity of wheelchair users [34]. The differences in motor skills or experience in wheelchair driving could lead to a different influence of the additional cognitive task. It would also be interesting to examine the differences between the genders more closely in order to determine whether, for example, different upper body strength has an influence on the observed changes.

## 5 Conclusion

Changes in spatiotemporal propulsion parameters observed during walking with an additional cognitive task were also observed when driving a manual wheelchair. This confirms that there is also an increase in the overall attention requirements and that the two tasks interfere. We observed a decline in the motor task while the cognitive task did not differ. In order to obtain more realistic results, measurements under dual-task conditions should be included in future movement analyses of people in manual wheelchairs. However, further studies with additional participants, tasks and environments for dual-task movement analysis of wheelchair users are required to further increase generalizability and closeness to reality.

## Supporting information

**S1 Table. Comparison of the mean spatiotemporal propulsion parameters variability during single- and dual-task driving a wheelchair.** Note: significant differences between manual wheelchair users and novices are marked by '*'. Due to the lack of a normal distribution, the Mann-Whitney U-test and Wilcoxon test were used to evaluate these parameters. (PDF)

## Acknowledgements

The present work is part of the doctoral thesis of Leon Salm and was performed in (partial) fulfillment of the requirements for obtaining the degree "Dr. med."

## Author contributions

**Conceptualization:** Leon Salm, Lucas Schreff, Roy Müller.

**Data curation:** Leon Salm, Lucas Schreff, Roy Müller.

**Formal analysis:** Leon Salm, Lucas Schreff, Roy Müller.

**Investigation:** Leon Salm, Lucas Schreff, Roy Müller.

**Methodology:** Leon Salm, Lucas Schreff, Roy Müller.

**Project administration:** Leon Salm, Roy Müller.

**Resources:** Leon Salm, Roy Müller.

**Software:** Lucas Schreff.

**Supervision:** Rainer Abel.

**Visualization:** Leon Salm, Lucas Schreff.

**Writing – original draft:** Leon Salm.

**Writing – review & editing:** Lucas Schreff, Christian Benker, Rainer Abel, Roy Müller.

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
