## [Decision Letter · Decision Letter 0]

25 Sep 2024

PONE-D-24-30440The effect of a cognitive dual-task on the control of wheelchair propulsionPLOS ONE

Dear Dr. Salm,

Thank you for submitting your manuscript to PLOS ONE. After careful consideration, we feel that it has merit but does not fully meet PLOS ONE’s publication criteria as it currently stands. Therefore, we invite you to submit a revised version of the manuscript that addresses the points raised during the review process.

We look forward to receiving your revised manuscript.

Kind regards,

Giancarlo Condello, Ph.D.

Academic Editor

PLOS ONE

**Journal Requirements:**

2. Please note that your Data Availability Statement is currently missing the repository name. If your manuscript is accepted for publication, you will be asked to provide these details on a very short timeline. We therefore suggest that you provide this information now, though we will not hold up the peer review process if you are unable.

Reviewers' comments:

Reviewer's Responses to Questions

**Comments to the Author**

1. Is the manuscript technically sound, and do the data support the conclusions?

Reviewer #1: Yes

Reviewer #2: Partly

2. Has the statistical analysis been performed appropriately and rigorously? 

Reviewer #1: Yes

Reviewer #2: Yes

3. Have the authors made all data underlying the findings in their manuscript fully available?

Reviewer #1: Yes

Reviewer #2: Yes

4. Is the manuscript presented in an intelligible fashion and written in standard English?

Reviewer #1: Yes

Reviewer #2: Yes

5. Review Comments to the Author

**Reviewer #1:**  The clearly written manuscript is highly relevant and presents important findings for wheelchair users and research that has been conducted in this field today generally ignoring the effect of cognitive dual-task on propulsion technique. Suggested modifications:

- Missing link to why this information is valuable and what is the impact of reduced performance in propulsion control, specifically I am missing the link to shoulder loads, injury, and pain.

- Furthermore, while the authors spend an extensive amount of time to compare the literature to cognitive dual-task on walking (I believe more then 50% of the references refers to walking), it is not clear to me why this comparison is valuable and what it adds? You could rather try to include more references to the cognitive task of wheelchair propulsion itself: e.g. https://www.ideals.illinois.edu/items/125182, https://ieeexplore.ieee.org/abstract/document/9086088

- In the abstract it is no clear how you defined ‘practiced’ and ‘unpracticed’. In general, suggest referring to e.g. manual wheelchair users and novices, instead.

- Line 36-43: this section is a bit unclear to me and difficult to follow. PASAT needs to be defined.

- Line 62-64: either explain the tests here or don’t go in so much detail and rather explain that multiple tests are being used… .

- Line 129: which marker set did you use?

- Line 162: what is this spoke-marker exactly, is it an IMU? Describe and explain.

- Line 262: the novice wheelchair users were already propelling with the most simplified / likely least cognitive demanding way of propelling offering less freedom to change? Experienced wheelchair users may also be more flexible to adapt their propulsion style depending on the cognitive requirement. Generally, as your data nicely shows, experienced wheelchair users apply longer, smoother strokes, as the cognitive requirements increases it is very interesting to see that their style is changing. This is a critical finding and to me most important.

- Line 285-287: again why is the comparison to walking valuable? What does it add, would sign shorten here.

- Line 337-338: This is a nice recommendation, you could make it stronger that your results with this new ‘marker’ follow the findings of e.g. push angle differences etc.

**Reviewer #2:**  Titled “The effect of a cognitive dual-task on the control of wheelchair propulsion”, the present study investigates the effects of dual-task conditions on wheelchair propulsion by adapting the dual-task methodology to wheelchair users. To examine how an additional cognitive task (the Paced Auditory Serial Addition Test, PASAT) affects wheelchair propulsion, 52 participants were divided into two groups: practiced (n=38) and unpracticed (n=14) wheelchair users. The key finding of the study is that dual-task conditions influence spatiotemporal propulsion parameters, with most significant changes observed in the practiced group.

Whilst the study strengths include the innovative approach (applying the dual-task paradigm, usually limited to walking studies, to wheelchair propulsion), use of advanced technology, clear division of groups, and novel methodology to quantify the distance between the hand and the pushrim, relevant limitations must be recognized since they may limit the generalizability of the results. In fact, the study weaknesses are the relatively small sample size for the unpracticed group (n=14), limiting the statistical power to detect significant differences in some propulsion parameters, lack of real-world conditions, gender imbalance (particularly in the practiced group: 27 males, 11 females), and the limited cognitive task variability. While the PASAT test is well-suited for assessing divided attention and working memory, using additional cognitive tasks could provide more comprehensive insights into how different types of cognitive loads (e.g., visual vs. auditory distractions) affect propulsion behavior.

The manuscript addresses an important and valuable topic, still there are certain inconsistencies that need to be resolved. Additionally, the research area lacks a comprehensive analysis of the literature. To enhance the rigor and quality of the study, specific revisions are required.

Overall comments:

1. The introduction needs to be strengthened.

2. Important statistical information is needed.

3. The discussion section has to be improved.

Specific comments:

Title

The study design can be included in the title.

Abstract

Please provide some basic demographic information such as means and standard deviation for the age and percentage of female participants (XX±YY years; F=ZZ%).

Please indicate the p values for the main reported statistical significances.

Introduction

Although the introduction is clear, it could be improved as follows:

1. Introducing general statistics on wheelchair users and the barriers they face could significantly enhance the introduction by highlighting the broader societal relevance of the study. These statistics provide context, establish the scale of the problem, and emphasize the importance of improving mobility solutions for wheelchair users.

2. Incorporating relevant studies that contextualize the current research examining movement dynamics and dual-task performance in other populations which often suffer movement limitations like older adults. See for instance: Ciaccioni, S., Capranica, L., Forte, R., Pesce, C., & Condello, G. (2020). Effects of a 4-month judo program on gait performance in older adults. The Journal of Sports Medicine and Physical Fitness, 60(5), 685-692.

3. Clarifying the research gap more explicitly.

Methods

Participants

The ethics review board approval date missing.

Please clarifying the study design (e.g., cross-sectional study) typology.

Procedures

Figure 1: clear.

Considering the mental arithmetic calculative nature of the PASAT test, have the authors asked the participants whether they had any issues with number-processing and calculation such as dyscalculia? For more information, see: Hiscock, M., Caroselli, J. S., & Kimball, L. E. (1998). Paced serial addition: Modality-specific and arithmetic-specific factors. Journal of clinical and experimental neuropsychology, 20(4), 463-472.

Data processing and statistical analysis

Please indicate whether there were missing data and how they have been eventually treated.

The authors should present their reasons for recruiting and selecting the number of people included and analyzed, for instance noting the statistical power of the study. A paragraph may explain the sample size needed to detect a hypothesized difference in outcomes.

To provide meaningful analysis for comparisons from small groups, please provide a measure of effect sizes (e.g., Cohen’s d).

Results

They are clear.

Tables 1 and 2

Clear

Discussion

Although the discussions are solid and well-organized, they sometimes use a too colloquial language (e.g., “If you see the recovery phase when driving a wheelchair…”). Moreover, a review of the literature is needed to compare the results with specific studies on sport and dual-tasks.

Please highlight better the limitations of the present research.

Suggestions for improvement and future studies should encompass: 1. Recruiting a larger and more balanced sample, particularly for the unpracticed group. A more diverse participant pool would allow for more detailed subgroup analyses, such as comparing users with different levels of experience or physical ability; 2. Incorporate real-world conditions, measuring wheelchair propulsion on different surfaces (e.g., pavement, grass) or in the presence of visual and auditory distractions to provide a more comprehensive understanding of how dual-task conditions affect wheelchair use in everyday life; 3. Broadening the cognitive task selection that simulate everyday distractions, such as talking on the phone or navigating visually busy environments, could yield more ecologically valid results; and 4. Considering gender differences, particularly analyzing whether gender-specific factors, such as upper body strength, impact propulsion under cognitive load.

Conclusions

Overall positive. Please revise it after improving the discussion section.

References

Please double-check the correctness of all references, according to the journal guidelines.

Please consider expanding the references after a specific review of the sport and dual task-related literature.

Supplementary Materials:

Clear.

6. PLOS authors have the option to publish the peer review history of their article (what does this mean? ). If published, this will include your full peer review and any attached files.

**Do you want your identity to be public for this peer review?** For information about this choice, including consent withdrawal, please see our Privacy Policy .

Reviewer #1: **Yes: ** Fransiska M Bossuyt

Reviewer #2: **Yes: ** Simone Ciaccioni

---

## [Author Response · Author response to Decision Letter 0]

4 Oct 2024

Dear reviewers,

thank you very much for your valuable and helpful comments that guided us in improving the manuscript.

Changed manuscript text is marked in blue in both the manuscript and our responses to your comments.

Kind regards

Leon Salm

Reviewer #1

The clearly written manuscript is highly relevant and presents important findings for

wheelchair users and research that has been conducted in this field today generally

ignoring the effect of cognitive dual-task on propulsion technique. Suggested

modifications:

- Missing link to why this information is valuable and what is the impact of reduced

performance in propulsion control, specifically I am missing the link to shoulder loads,

injury, and pain.

Response: You are right, some introductory information on the relevance of healthy

biomechanics when driving manual wheelchairs should be provided. We have added

this to the introduction.

- Furthermore, while the authors spend an extensive amount of time to compare the

literature to cognitive dual-task on walking (I believe more then 50% of the references

refers to walking), it is not clear to me why this comparison is valuable and what it

adds? You could rather try to include more references to the cognitive task of

wheelchair propulsion itself: e.g. https://www.ideals.illinois.edu/items/125182,

https://ieeexplore.ieee.org/abstract/document/9086088

Response: The reason why we compare our results with the results for walking and

also refer to the corresponding literature on dual-task walking is that there are no

studies on cognitive dual-task wheelchair driving apart from the dissertation you

suggested, which was also not accessible on request. Therefore, and this was the aim

of our study, we transferred the procedure from walking to wheelchair driving in order

to investigate whether the changes under dual-task conditions represent a universal

movement principle and whether the changes are similar in wheelchair driving. To

make this clearer, we have accentuated this aim in the introduction. Furthermore, we

also referred to the high impact of wheelchair use on cognitive and physical workload

in the introduction to mention these findings as well.

- In the abstract it is no clear how you defined ‘practiced’ and ‘unpracticed’. In general,

suggest referring to e.g. manual wheelchair users and novices, instead.

Response: Thank you for this suggestion, which we have implemented as proposed.

- Line 36-43: this section is a bit unclear to me and difficult to follow. PASAT needs to

be defined.

Response: We would also like to thank you for this comment. In this section we have

tried to present the experimental setup as briefly and clearly as possible. A more

detailed description would unfortunately go beyond the scope of an abstract. To make

it easier to understand, however, we have written out the Paced Auditory Serial

Addition Test instead of the acronym.

- Line 62-64: either explain the tests here or don’t go in so much detail and rather

explain that multiple tests are being used… .

Response: In accordance with your suggestion, we have made this part more general

and thus clearer.

- Line 129: which marker set did you use?

Response: We did not use a specific marker set, but were guided by experimental

setups from previous studies that performed marker-based 3d motion analyses of

manual wheelchairs. Please see e.g. https://pubmed.ncbi.nlm.nih.gov/26674751/

- Line 162: what is this spoke-marker exactly, is it an IMU? Describe and explain.

Response: The acceleration and deceleration of the rear wheel was determined via

double derivation of the path of the motion-capturing marker attached to the spokes.

We have rewritten the passage to make it clearer.

- Line 262: the novice wheelchair users were already propelling with the most simplified

/ likely least cognitive demanding way of propelling offering less freedom to change?

Experienced wheelchair users may also be more flexible to adapt their propulsion style

depending on the cognitive requirement. Generally, as your data nicely shows,

experienced wheelchair users apply longer, smoother strokes, as the cognitive

requirements increases it is very interesting to see that their style is changing. This is

a critical finding and to me most important.

Response: Thank you very much for these valuable considerations. We totally agree.

We have added these theories in the related paragraph.

- Line 285-287: again why is the comparison to walking valuable? What does it add,

would sign shorten here.

Response: Please see response to second comment above.

- Line 337-338: This is a nice recommendation, you could make it stronger that your

results with this new ‘marker’ follow the findings of e.g. push angle differences etc.

Response: If we understand you correctly, we very much agree that it is very

interesting to compare our new method with previous ones. Even if this is not yet

possible with this study, we are planning to compare this method with previous

approaches, such as the SmartWheel, in future studies.

Reviewer #2

Titled “The effect of a cognitive dual-task on the control of wheelchair propulsion”, the

present study investigates the effects of dual-task conditions on wheelchair propulsion

by adapting the dual-task methodology to wheelchair users. To examine how an

additional cognitive task (the Paced Auditory Serial Addition Test, PASAT) affects

wheelchair propulsion, 52 participants were divided into two groups: practiced (n=38)

and unpracticed (n=14) wheelchair users. The key finding of the study is that dual-task

conditions influence spatiotemporal propulsion parameters, with most significant

changes observed in the practiced group.

Whilst the study strengths include the innovative approach (applying the dual-task

paradigm, usually limited to walking studies, to wheelchair propulsion), use of

advanced technology, clear division of groups, and novel methodology to quantify the

distance between the hand and the pushrim, relevant limitations must be recognized

since they may limit the generalizability of the results. In fact, the study weaknesses

are the relatively small sample size for the unpracticed group (n=14), limiting the

statistical power to detect significant differences in some propulsion parameters, lack

of real-world conditions, gender imbalance (particularly in the practiced group: 27

males, 11 females), and the limited cognitive task variability. While the PASAT test is

well-suited for assessing divided attention and working memory, using additional

cognitive tasks could provide more comprehensive insights into how different types of

cognitive loads (e.g., visual vs. auditory distractions) affect propulsion behavior.

The manuscript addresses an important and valuable topic, still there are certain

inconsistencies that need to be resolved. Additionally, the research area lacks a

comprehensive analysis of the literature. To enhance the rigor and quality of the study,

specific revisions are required.

Overall comments:

1. The introduction needs to be strengthened.

2. Important statistical information is needed.

3. The discussion section has to be improved.

Response: Done as suggested. Please see response to specific comments below.

Specific comments:

Title

The study design can be included in the title.

Response: We would like to thank you for this suggestion and have added the study

design to the methods section, as described below. However, for the sake of

readability, we have decided against including the study design in the title.

Abstract

Please provide some basic demographic information such as means and standard

deviation for the age and percentage of female participants (XX±YY years; F=ZZ%).

Response: Thank you very much for this recommendation, which we have

implemented as suggested.

Please indicate the p values for the main reported statistical significances.

Response: We also implemented this recommendation as suggested.

Introduction

Although the introduction is clear, it could be improved as follows:

1. Introducing general statistics on wheelchair users and the barriers they face could

significantly enhance the introduction by highlighting the broader societal relevance of

the study. These statistics provide context, establish the scale of the problem, and

emphasize the importance of improving mobility solutions for wheelchair users.

2. Incorporating relevant studies that contextualize the current research examining

movement dynamics and dual-task performance in other populations which often suffer

movement limitations like older adults. See for instance: Ciaccioni, S., Capranica, L.,

Forte, R., Pesce, C., & Condello, G. (2020). Effects of a 4-month judo program on gait

performance in older adults. The Journal of Sports Medicine and Physical Fitness,

60(5), 685-692.

3. Clarifying the research gap more explicitly.

Response: Thank you for this suggestion. We have revised the introduction and

highlighted the high relevance of our study in a new paragraph, especially regarding

the ongoing problem of upper extremity pain and injury. This background information

was missing before, but now the importance should be made clear. In our detailed

presentation of the findings obtained from dual-task movement analysis on walking,

we also cited studies that examined older adults or even investigated the changes in

relation to age. Please see e.g. https://pubmed.ncbi.nlm.nih.gov/16541455/. Your

study on the effect of a 4-month judo program on the gait performance of older adults

is very interesting, but unfortunately not quite suitable for a comparison with our study.

However, we will keep it in mind for future publications.

Methods

Participants

The ethics review board approval date missing.

Response: Thank you for noticing. We have added the date accordingly.

Please clarifying the study design (e.g., cross-sectional study) typology.

Response: Following your suggestion, we have added to the methods section that the

study is a cross-sectional study.

Procedures

Considering the mental arithmetic calculative nature of the PASAT test, have the

authors asked the participants whether they had any issues with number-processing

and calculation such as dyscalculia? For more information, see: Hiscock, M., Caroselli,

J. S., & Kimball, L. E. (1998). Paced serial addition: Modality-specific and arithmetic-

specific factors. Journal of clinical and experimental neuropsychology, 20(4), 463-472.

Response: Thank you very much for this valuable note. The additional use of other

tasks for the dual-task conditions would have been an improved way to recognize the

interference from individual difficulties. We have included this concern in the

presentation of limitations.

Data processing and statistical analysis

Please indicate whether there were missing data and how they have been eventually

treated.

Response: Small gaps in the records of the motion-capturing markers were filled using

the gap filler function in our recording program Nexus. Otherwise, no data was missing

for any of the included participants.

The authors should present their reasons for recruiting and selecting the number of

people included and analyzed, for instance noting the statistical power of the study. A

paragraph may explain the sample size needed to detect a hypothesized difference in

outcomes.

Response: Thanks for this advice. We calculated the sample size using an a priori

power analysis for an ANOVA Repeated measures model conducted by means of

G*Power 3.1.5 software, given the following input parameters: effect size: F = 0.2;

alpha error probability: 0.05; power: 0.8. However, as it was not possible for us to

obtain subgroups of equal size, this represents a limitation of the statistical power,

which we have included in the limitations.

To provide meaningful analysis for comparisons from small groups, please provide a

measure of effect sizes (e.g., Cohen’s d).

Response: We have included the effect sizes of the differences between the two

groups using Cohen's d in the limitations

Discussion

Although the discussions are solid and well-organized, they sometimes use a too

colloquial language (e.g., “If you see the recovery phase when driving a wheelchair…”).

Moreover, a review of the literature is needed to compare the results with specific

studies on sport and dual-tasks.

Response: Thank you for this notice. We have adjusted the language level at the

required passages. Even though the findings of dual-task movement analyses in sports

are very interesting, they are not particularly suitable for comparison with our results,

which is why we have decided not to discuss them in detail.

Please highlight better the limitations of the present research.

Response: This suggestion is very reasonable. We have revised the discussion and

added a new section describing the limitations, in which they are clearly presented.

Suggestions for improvement and future studies should encompass: 1. Recruiting a

larger and more balanced sample, particularly for the unpracticed group. A more

diverse participant pool would allow for more detailed subgroup analyses, such as

comparing users with different levels of experience or physical ability; 2. Incorporate

real-world conditions, measuring wheelchair propulsion on different surfaces (e.g.,

pavement, grass) or in the presence of visual and auditory distractions to provide a

more comprehensive understanding of how dual-task conditions affect wheelchair use

in everyday life; 3. Broadening the cognitive task selection that simulate everyday

distractions, such as talking on the phone or navigating visually busy environments,

could yield more ecologically valid results; and 4. Considering gender differences,

particularly analyzing whether gender-specific factors, such as upper body strength,

impact propulsion under cognitive load.

Response: Thank you very much for these great ideas. We have included your

suggestions in the revised discussion.

Conclusions

Overall positive. Please revise it after improving the discussion section.

Response: We have revised the conclusion as proposed.

References

Please double-check the correctness of all references, according to the journal

guidelines.

Please consider expanding the references after a specific review of the sport and dual

task-related literature.

Response: We have once again checked the references and expanded them to

include the newly added sources.

---

## [Decision Letter · Decision Letter 1]

31 Dec 2024

The effect of a cognitive dual-task on the control of wheelchair propulsion

PONE-D-24-30440R1

Dear Dr. Salm,

We’re pleased to inform you that your manuscript has been judged scientifically suitable for publication and will be formally accepted for publication once it meets all outstanding technical requirements.

Kind regards,

Giancarlo Condello, Ph.D.

Academic Editor

PLOS ONE

Additional Editor Comments (optional): NA

Reviewers' comments:

Reviewer's Responses to Questions

**Comments to the Author**

1. If the authors have adequately addressed your comments raised in a previous round of review and you feel that this manuscript is now acceptable for publication, you may indicate that here to bypass the “Comments to the Author” section, enter your conflict of interest statement in the “Confidential to Editor” section, and submit your "Accept" recommendation.

Reviewer #1: All comments have been addressed

Reviewer #2: All comments have been addressed

2. Is the manuscript technically sound, and do the data support the conclusions?

Reviewer #1: Yes

Reviewer #2: (No Response)

3. Has the statistical analysis been performed appropriately and rigorously? 

Reviewer #1: Yes

Reviewer #2: (No Response)

4. Have the authors made all data underlying the findings in their manuscript fully available?

Reviewer #1: Yes

Reviewer #2: (No Response)

5. Is the manuscript presented in an intelligible fashion and written in standard English?

Reviewer #1: Yes

Reviewer #2: (No Response)

6. Review Comments to the Author

Reviewer #1: The authors carefully responded to each of the reviewers' comments and clarity of the manuscript further improved. I very much appreciate the modifications and enjoyed reading the revised manuscript. I believe that the only thing I am missing is that although the authors carefully described how optimal propulsion biomechanics include long and smooth strokes, there could be a couple of words added to the discussion stating that this is not only associated with efficiency but also with reduced shoulder loads and therefore is included in the guidelines to preserve shoulder health in wheelchair users. Changes to propelling with a smaller push angle have also been observed as a compensation mechanism in response to fatiguing propulsion in wheelchair users. It therefore appears that when pushing the system (either through fatigue or dual-task performance), wheelchair users compensate for the 'easier' propulsion style, yet this may increase demands on shoulder and potentially lead to injury. Thank you for your work.

Reviewer #2: (No Response)

7. PLOS authors have the option to publish the peer review history of their article (what does this mean? ). If published, this will include your full peer review and any attached files.

**Do you want your identity to be public for this peer review?** For information about this choice, including consent withdrawal, please see our Privacy Policy .

Reviewer #1: **Yes: ** Fransiska M Bossuyt

Reviewer #2: No

---

## [Editor Report · Acceptance letter]

PONE-D-24-30440R1

PLOS ONE

Dear Dr. Salm,

I'm pleased to inform you that your manuscript has been deemed suitable for publication in PLOS ONE. Congratulations! Your manuscript is now being handed over to our production team.

Kind regards,

on behalf of

Dr. Giancarlo Condello

Academic Editor

PLOS ONE